# Kinetics of Water-Induced Amorphous Phase Separation in Amorphous Solid Dispersions via Raman Mapping

**DOI:** 10.3390/pharmaceutics15051395

**Published:** 2023-05-02

**Authors:** Adrian Krummnow, Andreas Danzer, Kristin Voges, Samuel O. Kyeremateng, Matthias Degenhardt, Gabriele Sadowski

**Affiliations:** 1Laboratory of Thermodynamics, Department of Biochemical and Chemical Engineering, TU Dortmund University, Emil-Figge-Street 70, D-44227 Dortmund, Germany; 2AbbVie Deutschland GmbH & Co. KG, Global Pharmaceutical R&D, Knollstraße, D-67061 Ludwigshafen am Rhein, Germany

**Keywords:** phase behavior, miscibility gap, amorphous phase separation, kinetics, Raman imaging, confocal Raman spectroscopy, copovidone

## Abstract

The poor bioavailability of an active pharmaceutical ingredient (API) can be enhanced by dissolving it in a polymeric matrix. This formulation strategy is commonly known as amorphous solid dispersion (ASD). API crystallization and/or amorphous phase separation can be detrimental to the bioavailability. Our previous work (*Pharmaceutics 2022, 14(9), 1904*) provided analysis of the thermodynamics underpinning the collapse of ritonavir (RIT) release from RIT/poly(vinylpyrrolidone-co-vinyl acetate) (PVPVA) ASDs due to water-induced amorphous phase separation. This work aimed for the first time to quantify the kinetics of water-induced amorphous phase separation in ASDs and the compositions of the two evolving amorphous phases. Investigations were performed via confocal Raman spectroscopy, and spectra were evaluated using so-called Indirect Hard Modeling. The kinetics of amorphous phase separation were quantified for 20 wt% and 25 wt% drug load (DL) RIT/PVPVA ASDs at 25 °C and 94% relative humidity (RH). The in situ measured compositions of the evolving phases showed excellent agreement with the ternary phase diagram of the RIT/PVPVA/water system predicted by PC-SAFT in our previous study (*Pharmaceutics 2022, 14(9), 1904*).

## 1. Introduction

Amorphous solid dispersions (ASDs) enhance the bioavailability of poorly water-soluble active pharmaceutical ingredients (APIs) [1,2,3]. In ASDs, the API is dissolved in a suitable polymer matrix. However, ASDs are often supersaturated regarding the solubility of the crystalline and/or the amorphous API in the polymer, eventually resulting in crystallization and/or amorphous phase separation, respectively. The loss of homogeneity in the solid state is highly unwanted and additionally promoted and accelerated by water sorption when exposing an ASD to humidity or to an aqueous dissolution medium [4,5,6,7,8,9,10]. Thus, the physical stability and release mechanism of ASDs might be strongly influenced by amorphous phase separation [11]. Therefore, investigating amorphous phase-separation kinetics in ASDs is of enormous interest [12,13,14,15,16,17,18,19,20,21,22].

Solely based on the phase equilibria, the kinetics of the amorphous phase separation into an amorphous API-rich and an amorphous API-poor phase cannot be predicted. However, the spatially resolved measurement of the phase-separation kinetics in ASDs is challenging due to the high viscosity of ASDs [23], which results in long equilibration times. Moreover, the experimental setups are complex, as the measurement technique (e.g., fluorescence, infrared, or Raman spectroscopy) has to offer a sufficient spatial resolution to distinguish between the continuous phase and the dispersed phase [24,25,26,27,28].

To investigate the kinetics of amorphous phase separation, it is necessary to prevent API crystallization during the measurement. Therefore, measurements must be performed at temperatures above the melting temperature of the ASD or for ASDs with a low API crystallization tendency. The first option was selected by Luebbert et al. [4,29] for a confocal Raman-spectroscopy experiment, whereby the Raman spectra served to determine the concentrations of every component in both phases using so-called Indirect Hard Modeling (IHM). Recurrent line mapping enabled an in situ quantification of the kinetics of amorphous phase separation in a dry ibuprofen/Resomer^®^ RG752S ASD with a 70 wt% drug load (DL) at 75 °C.

The other option was chosen by Shi et al. [30] considering ASDs composed of ritonavir (RIT) and poly(vinylpyrrolidone-co-vinyl acetate) (PVPVA) due to the very low crystallization tendency of RIT [31]. They performed X-ray fluorescence experiments to determine the onset of water-induced amorphous phase separation in situ using recurrent maps in RIT/PVPVA ASDs. The spectrum contributions of sulfur atoms originating from RIT were used to distinguish between the RIT-rich and the RIT-poor phases without quantifying their compositions. These measurements proved that the kinetics of amorphous phase separation increases with increasing DL, leading to the switch in the release mechanism during dissolution (from congruent to non-congruent), as rigorously explained in our previous work [11]. The influence of the four surfactants sodium dodecyl sulfate (SDS), polyoxyethylene sorbitan monooleate (Tween^®^ 80), sorbitan monolaurate (Span^®^ 20), and sorbitan trioleate (Span^®^ 85) on water-induced amorphous phase separation in RIT/PVPVA ASDs at 37 °C was investigated in situ by Yang et al. [32] via confocal fluorescence microscopy. They exploited the different partitioning of Nile red and Alexa Fluor^®^ 488 into the two phases to distinguish the RIT-rich and RIT-poor phase but did not quantify the phase compositions. The resulting microstructures were strongly dependent on the present surfactant and showed significant differences from the neat case.

The only work which quantified the kinetics of amorphous phase separation in an ASD was authored by Luebbert et al. [29] but in a dry system. Other working groups only report the descriptive mechanism of amorphous phase separation and qualitative compositions of the evolved phases. This work aims for the first time to quantify the kinetics of water-induced amorphous phase separation in ASDs. Due to the outstanding significance of phase-separation kinetics in understanding and explaining the release mechanism of RIT/PVPVA ASD, it was chosen as the model system for this study [11]. Confocal Raman spectroscopy was applied to quantify the kinetics of amorphous phase separation in ASDs with DLs of 20 wt% and 25 wt% at 25 °C and 94% relative humidity (RH). Gravimetric water-sorption measurements additionally validated the spectra evaluation of ternary mixtures via so-called Indirect Hard Modeling. The compositions in the evolving phases were used to perfectly validate the miscibility gap in the RIT/PVPVA/water ternary phase diagram as earlier predicted by the Perturbed-Chain Associating Fluid Theory (PC-SAFT) [11].

## 2. Materials and Methods

### 2.1. Materials

RIT (melting temperature of 124.97 °C and degradation temperature of 160 °C [33,34]) was supplied by AbbVie Deutschland GmbH & Co. KG (Ludwigshafen, Germany). The copolymer PVPVA (Kollidon^®^ VA64) with a weight-average molar mass of 65,000 g/mol was obtained from BASF SE (Ludwigshafen, Germany). Silica gel was purchased from BINDER GmbH (Tuttlingen, Germany). Sodium nitrate (purity ≥ 99%) was purchased from Merck KGaA (Darmstadt, Germany), and potassium nitrate (purity ≥ 99%) was purchased from VWR International GmbH (Darmstadt, Germany). All substances were used without further purification. Water was purified using a Milli-Q^®^ Advantage A10 purification system of Merck KGaA (Darmstadt, Germany).

### 2.2. Methods

#### 2.2.1. Ball Milling

Prior to ASD manufacturing (DLs of 0 wt%, 10 wt%, 20 wt%, 70 wt%, and 100 wt%) via vacuum compression molding, physical mixtures (a total of 0.7 g of RIT and PVPVA in respective amounts) were prepared using a ball mill Pulverisette 23 by Fritsch GmbH (Idar-Oberstein, Germany). A 10 mL stainless-steel cup and a ⌀15 mm stainless-steel ball were chosen for milling. The latter was applied three times for 3 min at 50 Hz with two breaks of 60 s for heat removal. Each physical mixture was stored in a vacuum chamber for at least 24 h before further usage.

#### 2.2.2. Vacuum Compression Molding (VCM)

ASDs for gravimetric water-sorption measurements and confocal Raman spectroscopy were manufactured using a ⌀8 mm VCM disc tool from MeltPrep GmbH (Graz, Austria). Depending on the DL, 0.018 g of a dried physical mixture was heated to an internal temperature of 140–150 °C within 7–20 min. To prevent material loss due to low melt viscosity, ASDs with DLs of 70 wt% and 100 wt% were produced without applying vacuum. For all other ASDs, vacuum was applied. In the subsequent cooling step, the ASDs were kept at 25 °C for 10 min. After production, the ASDs were stored immediately at the desired conditions (see Section 2.2.3, Section 2.2.4 and Section 2.2.5).

#### 2.2.3. Gravimetric Water-Sorption Measurements

To measure samples stored at 25 °C at constant humid conditions, saturated salt solutions were prepared with an excess of sodium nitrate (74% RH) and potassium nitrate (94% RH) [35]. The ASDs were weighed before and after at least two days of storage in a climate chamber from BINDER GmbH (Tuttlingen, Germany) at 25 °C and fixed RH (isopiestic method) to determine the mass increase (accuracy ± 0.005 mg) due to water sorption. Thermodynamic equilibrium was assumed when the mass remained constant. We always ensured via modulated differential scanning calorimetry (procedure analogous to our previous publication [11]) that measurements were not influenced by recrystallization of RIT.

#### 2.2.4. Confocal Raman Spectroscopy

Raman experiments were conducted to verify the homogeneity of freshly prepared ASDs and to measure the kinetics of amorphous phase separation in ASDs at 25 °C and 94% RH. The measurements were performed using a LabRAM 300 Raman spectroscope from HORIBA Jobin Yvon GmbH (Bensheim, Germany) coupled with an inverted microscope IX71 from Olympus Corp. (Tokyo, Japan). The resulting experimental setup was operated via the software LabSpec 6 from HORIBA Jobin Yvon GmbH (Bensheim, Germany) and is schematically shown in Figure 1 with its relevant parts.

A Torus laser system from Laser Quantum Ltd. (Stockport, UK) emitted light with a wavelength of 532 nm and a power of 300 mW. A neutral-density filter wheel was set to a transmittance of 100% for calibration and verification of ASD homogeneity and to a transmittance of 25% for measurements regarding amorphous phase separation in ASDs. Thus, during the latter measurements, the initial laser intensity was reduced by 75% to suppress thermal and photochemical degradation of the samples due to the focused laser spot. Focusing the laser and collecting the backscattering was achieved via an infinity-corrected LMPlanFl 100×/0.8 objective from Olympus Corp. (Tokyo, Japan). A notch filter was used as a dichroic mirror to reflect the laser and to transmit the Raman scattering. The latter passed a confocal pinhole and a slit adjusted to 1000 µm and 500 µm, respectively. The beam was dispersed by a 600 mm^−1^ diffraction grating and projected onto a Peltier-cooled charge-coupled device (CCD). The CCD had a size of 1024 × 56 pixels and a dynamic range of 16 bits (pixel size of 27 µm) from Andor Technology Ltd. (Belfast, UK). The central spectrograph position was 2249.74 cm^−1^, and the wavenumber was calibrated with pure crystalline silicon. All spectra were recorded with an integration time of 1 s.

The sealed measurement cell (vapor-phase volume of 1.5 mL) had a quartz-glass cover lid at the bottom, and the cell temperature was controlled to 25 °C (accuracy ± 1 K). Moreover, the entire cell was placed on a movable, motorized XY stage SCAN IM 120 × 80 from Märzhäuser Wetzlar GmbH & Co. KG (Wetzlar, Germany) controlled by a TANGO Desktop from Märzhäuser Sensotech GmbH (Wetzlar, Germany). Depending on the sample, three different measurement cell setups were used: Figure 1a–c show the setup for measuring aqueous solutions (low vapor phase volume), dry ASDs (silica gel as desiccator), and ASDs at fixed RH (isopiestic method), respectively. For the isopiestic method, saturated salt solutions were analogously prepared as described in Section 2.2.3.

We always ensured that measurements were not influenced by recrystallization of RIT via microscopy assessment and via comparing the measured mixture spectra to the crystalline spectrum of RIT [36].

#### 2.2.5. Calibration Model

Raman spectra were evaluated using so-called Indirect Hard Modeling (IHM) as developed by Alsmeyer et al. [37,38]. First, the pure-component spectra were represented by a specific number of superposed pseudo-Voigt functions (a pseudo-Voigt function is a linear combination of a Gaussian and a Lorentzian function). Then, shape, shifting, and weighting parameters were inversely determined to approximate mixture spectra of known compositions by a weighted sum of the pure-component spectra [39,40]. Parameter determination was carried out at the constraint of mass conservation ($\sum w_{i}$ = 1). The shape, shifting, and weighting parameters account for changing peak shapes as well as for nonlinear peak shifts and enable the reliable quantification of mixture spectra of unknown compositions even outside the calibrated composition range [41,42]. IHM was conducted applying the software PEAXACT 5.4 from S-PACT GmbH (Aachen, Germany).

Calibration spectra were recorded according to Appendix A for pure amorphous RIT, PVPVA, and water, as well as for binary, single-phase mixtures of RIT/PVPVA and PVPVA/water. Each calibration spectrum is an accumulation of 100 spectra recorded at a single point without moving the measurement cell. In case of measuring wet PVPVA after storage at 74% and 94% RH, the initially dry PVPVA was stored inside the isopiestic measurement cell for two days before recording the spectra.

## 3. Results and Discussion

### 3.1. Calibration Model

All pure-component spectra (Appendix A) were modeled using a superposition of pseudo-Voigt functions. The number of functions required reflects the complexity of the molecular structure of each pure component. Consequently, the spectra of amorphous RIT, PVPVA, and water were described with 32, 17, and 5 pseudo-Voigt functions, respectively.

Combining the modeled pure-component spectra and the recorded mixture spectra of known compositions enabled the determination of shape, shifting, and weighting parameters. This procedure resulted in the measured vs. true plot for the mass fractions of every component i, as depicted in Figure 2. The calibration model precisely describes RIT ($R^{2}$ = 0.9994), PVPVA ($R^{2}$ = 0.9992), and water ($R^{2}$ = 0.9996) over the entire concentration range. For additional statistical indicators underlining the high performance of our calibration model, the reader is referred to Appendix A.

### 3.2. Homogeneity of Freshly Prepared ASDs

Line mapping was performed using the setup from Figure 1 to prove the homogeneity of the freshly prepared ASDs. The measurement cell was moved point-by-point over 1000 µm using a step size of ΔX = 20 µm (51 points in total). At each point, 100 recorded spectra were accumulated. Appendix A shows the mass fractions of RIT and PVPVA as a function of the spatial coordinate X for ASDs with DLs of 20 wt% and 25 wt%. It was ensured that all measurements shown in this work started from homogeneous ASDs.

### 3.3. Raman Mapping of Amorphous Phase Separation

Mapping was conducted with the setup shown in Figure 1 to measure the kinetics of amorphous phase separation in RIT/PVPVA ASDs at 25 °C and 94% RH. The timeline started (t = 0) when the dry ASDs were placed in the measurement cell. Recording spectra started after 24 h for ASDs with 20 wt% DL and after 12 h for ASDs with 25 wt% DL to ensure enough sorption of water. Water has a high heat capacity of 4.2 kJ/(kg K) at 25 °C, resulting in negligible temperature increases in the wet ASD by the focused laser spot.

The measurement cell was moved point-by-point in X-direction and row-by-row in Y-direction over a single area of 50 µm × 50 µm for 20 wt% DL and 75 µm × 75 µm for 25 wt% DL. The step sizes were set to ΔX = ΔY = 1 µm leading to a total of 2601 points for 20 wt% DL and 5776 points for 25 wt% DL. The spectrum at each point is an accumulation of three recorded spectra to increase the signal-to-noise ratio. After one run lasting 2.43 h for 20 wt% DL and 5.39 h for 25 wt% DL, the measurement cell was returned to the starting position to measure the same area again to access the time evolution of concentration for each point. One hundred twenty-six runs were carried out for 20 wt% DL and 39 runs for 25 wt% DL, resulting in a total of 327,726 spectra and 225,264 spectra, respectively. Each run generated one map, and the spectra were evaluated via IHM to identify the spatial distribution of all components resulting in spatially resolved concentration maps. The time assigned to each map corresponds to the time at which the spectrum at the center point of each map was recorded.

#### 3.3.1. Amorphous Phase Separation in the ASD with 20 wt% DL

Figure 3 illustrates the spatial distributions of RIT, PVPVA, and water in the ASD with 20 wt% DL at 25 °C and 94% RH after 25.2 h (Figure 3a,d,g), 184.3 h (Figure 3b,e,h), and 330.2 h (Figure 3c,f,i). The timeline started (t = 0) with placing the dry ASD in the measurement cell, and the assigned times correspond to the times at which spectra were recorded at the map center points.

The depicted distribution of water at 25.2 h (Figure 3g) results in an average mass fraction of $w_{\mathrm{water}}$ = 0.26 ± 0.01. The value is also in very good agreement with the mass fraction $w_{\mathrm{water}}$ = 0.25 determined via gravimetric water-sorption measurements after storage at 25 °C and 94% RH [11], thus perfectly validating our calibration model. After 25.2 h, each component was homogeneously distributed, and the averaged composition of the wet ASD was $w_{\mathrm{RIT}}$ = 0.17 ± 0.03, $w_{\mathrm{PVPVA}}$ = 0.57 ± 0.02, and $w_{\mathrm{water}}$ = 0.26 ± 0.01.

The spatial component distributions at 184.3 h (Figure 3b,e,h) and 330.2 h (Figure 3c,f,i) show the formation of RIT-rich droplets dispersed in a continuous RIT-poor phase. The latter has a high polymer content ($w_{\mathrm{PVPVA}}^{L1}$ ≈ 0.60) resulting in high viscosity inhibiting mass transfer and thus coalescence of RIT-rich droplets [43,44]. The diameter of the RIT-rich droplets increased, and the composition in both phases evolved over time. At 184.3 h (Figure 3b,e,h) and 330.2 h (Figure 3c,f,i), the droplets had diameter ranges of approximately 2 µm to 5 µm and 3 µm to 7 µm, respectively.

Figure 4a,b depict the spectra corresponding to the RIT-poorest and RIT-richest points at 330.2 h (Figure 3c,f,i), respectively. The RIT-specific contributions to the spectra are particularly noticeable within the shaded Raman shift intervals in Figure 4. Rigorous evaluation of the two spectra via IHM led to the compositions of $w_{\mathrm{RIT}}^{L1}$ = 0.08, $w_{\mathrm{PVPVA}}^{L1}$ = 0.63, and $w_{\mathrm{water}}^{L1}$ = 0.29 for the RIT-poor phase L1 and $w_{\mathrm{RIT}}^{L2}$ = 0.46, $w_{\mathrm{PVPVA}}^{L2}$ = 0.36, and $w_{\mathrm{water}}^{L2}$ = 0.18 for the RIT-rich phase L2.

The findings above are typical for amorphous phase separation as a process that jeopardizes the homogeneity of ASDs. However, it was not possible to determine an onset of amorphous phase separation because the evolution of the two distinct coexisting phases was slow and continuous, and the spatial resolution limitation was in the micron range rather than in the nano range. Our results agree with observations from confocal fluorescence microscope images for a 20 wt% DL RIT/PVPVA ASD at 37 °C and RH → 100% by Yang et al. [32]. They reported that amorphous phase separation was only visible after longer exposure times. (We do not adopt the notation RH = 100% from the literature. In thermodynamic equilibrium, the water activity in the ASD has to be equal to the RH of the surrounding phase. Thus, RH = 100% would require the water activity in the ASD being one which can only be realized for pure (liquid) water. As soon as any other substance, e.g., polymer or API, is present, the water activity is always lower than one, which means that RH in the surrounding phase is always below 100%, which is achieved by dissolution of polymer and/or API.)

No recrystallization of RIT was observed during the conducted Raman mapping and is further supported by the investigation of Shi et al. [30]. They used an even higher RH (use of a water drop for RH → 100%) and did not detect any crystals via polarized light microscopy even after 168 h of storage.

#### 3.3.2. Amorphous Phase Separation in the ASD with 25 wt% DL

Figure 5 shows the spatial distributions of RIT, PVPVA, and water in the ASD with 25 wt% DL at 25 °C and 94% RH after 25.4 h (Figure 5a,d,g), 30.8 h (Figure 5b,e,h), and 133.6 h (Figure 5c,f,i). Again, the timeline started (t = 0) with placing the dry ASD in the measurement cell, and the assigned times correspond to the times at which spectra were recorded at the map center points.

The averaged mass fraction of $w_{\mathrm{water}}$ = 0.26 ± 0.02 at 25.4 h (Figure 5g) measured via Raman spectroscopy again supports our calibration model because the gravimetrically determined mass fraction in this work after storage at 25 °C and 94% RH was $w_{\mathrm{water}}$ = 0.25. A comparison of the spatial distributions at different times showed a completely different behavior compared with the previous 20 wt% DL ASD. Until 25.4 h (Figure 5a,d,g), each component was homogeneously distributed ($w_{\mathrm{RIT}}$ = 0.21 ± 0.04, $w_{\mathrm{PVPVA}}$ = 0.52 ± 0.03, and $w_{\mathrm{water}}$ = 0.26 ± 0.02 at 25.4 h); however, during the next mapping run (each run lasted 5.4 h so that no information was lost) at 30.8 h (Figure 5b,e,h), RIT-rich domains with a diameter range of approximately 5 µm to 25 µm dispersed in a continuous RIT-poor phase could be identified. The formation of these RIT-rich domains can be attributed to an initially enormous driving force (chemical-potential gradient) for phase separation (see also Section 3.4). Comparing the spatial distributions for 25.4 h (Figure 5a,d,g) and 30.8 h (Figure 5b,e,h) indicates that the onset for amorphous phase separation in the ASD with 25 wt% DL was between 25.4 h and 30.8 h, which was about 5.4 h after the equilibrium water content had been attained.

After 133.6 h (Figure 5c,f,i), the droplets were spherical and with diameters between 5 µm and 35 µm. Simultaneously, the compositions in both phases evolved remarkably between 30.8 h (Figure 5b,e,h) and 133.6 h (Figure 5c,f,i). Figure 6a,b show the spectra recorded from the RIT-poorest and RIT-richest point at 133.6 h, respectively. The spectrum of the RIT-poor phase L1 corresponds to a composition of $w_{\mathrm{RIT}}^{L1}$ = 0.01, $w_{\mathrm{PVPVA}}^{L1}$ = 0.67, and $w_{\mathrm{water}}^{L1}$ = 0.32 (Figure 6a). In contrast, the spectrum in Figure 6b reveals a composition of the evolving RIT-rich phase L2 of $w_{\mathrm{RIT}}^{L2}$ = 0.83, $w_{\mathrm{PVPVA}}^{L2}$ = 0.11, and $w_{\mathrm{water}}^{L2}$ = 0.06.

As can be seen by comparing Figure 5 to Figure 3, there were not only more diverging compositions of the two phases but also fewer and larger droplets in the ASD with 25 wt% DL compared to the ASD with 20 wt% DL. This indicates that the kinetics of amorphous phase separation in 25 wt% DL ASD was faster than the one in the 20 wt% DL ASD. Coarsening of the droplets can be attributed to coalescence and Ostwald ripening [45,46]. Big droplets grew at the expense of smaller ones to minimize the total Gibbs energy of the system by reducing the interfacial area between droplets and the continuous polymer-rich phase (e.g., see Figure 5b,c). The latter surrounds the RIT-rich droplets and slows down coalescence and Ostwald ripening as it acts as a highly viscous polymer-rich mass-transport barrier. Figure 7 depicts the mass fractions of all components along Y = 25 µm after 133.6 h corresponding to Figure 5c,f,i and showing this barrier, for example, at X = 42 µm. Despite these barriers, from a thermodynamic point of view, only two separated bulk phases will remain in the ASD at equilibrium after an infinite time.

### 3.4. Amorphous Phase Separation in the Ternary Phase Diagram

In our previous work [11], the ternary phase diagram of RIT/PVPVA/water at 25 °C and 0.1 MPa was predicted using PC-SAFT. The calculated miscibility gap was validated via multiple methods, such as RIT/PVPVA ASD dissolution experiments, as well as visual turbidity inspection and glass-transition temperature measurements after storing RIT/PVPVA ASDs at 25 °C and various RHs. A newly developed approach enabled the reliable determination of the compositions of the coexisting amorphous phases based on their measured glass-transition temperatures, the Kwei equation, and the law of mass conservation. This approach is the first-ever quantitative experimental validation for the predicted equilibrium compositions and phase mass ratios in RIT/PVPVA ASDs with 10 wt%, 20 wt%, 30 wt%, and 40 wt% DL at 25 °C and 94% RH. In this work, we applied Raman mapping to quantify the compositions of both phases over time, which can subsequently be applied to validate the PC-SAFT prediction in detail.

Figure 8 shows the ternary phase diagram of RIT/PVPVA/water at 25 °C and 0.1 MPa from our previous work, with the miscibility gap predicted by PC-SAFT and the glass-transition line calculated using the Kwei equation [11]. When the two initially dry (a) 20 wt% and (b) 25 wt% DL ASDs were exposed to water vapor, they absorbed water until the water activity in the wet ASD equaled the water activity (= RH) in the surrounding vapor phase. The resulting overall equilibrium water contents of the wet ASDs, gravimetrically determined in [11] and in this work, indicate the feed compositions for amorphous phase separation. Of course, this is a simplified point of view because wet ASD is prone to amorphous phase separation as soon as the miscibility gap is entered. However, this little inaccuracy should be negligible, as the kinetics of amorphous phase separation at low water contents is very slow compared to the kinetics of water sorption at 25 °C and 94% RH (as shown above).

The data points shown in Figure 8 were determined from the spatial distributions in Figure 3c,f,i at 330.2 h and Figure 5c,f,i at 133.6 h, for (a) 20 wt% DL and (b) 25 wt% DL, respectively. The compositional scattering of the data points can be attributed to existing concentration gradients during progressive amorphous phase separation. Comparing the determined compositions with the predicted tie-line slope reveals an excellent agreement between the prediction and experimental data for both DLs. Interestingly, the glassy region had no influence on the amorphous phase separation as all measured compositions are located in the rubbery region. The phase compositions resulting for the ASD with 25 wt% DL (Figure 8b) were found to be closer to approaching the predicted equilibrium compositions than the phase compositions in the ASD with 20 wt% DL, although the former was measured at one third the time scale of the former. This indicates that the kinetics of amorphous phase separation in the 25 wt% DL ASD is much faster than in the 20 wt% DL ASD. Thermodynamically, the phase compositions in both wet ASDs are expected to agree with the predicted equilibrium compositions at the end of the tie lines after an infinite time.

### 3.5. Kinetics of Amorphous Phase Separation

The preparation method of an ASD does not influence the thermodynamic equilibrium of phase separation, whereas it does influence its kinetics. For example, the properties of glasses are markedly affected by their preparation temperatures. Freshly prepared glasses are in a non-equilibrium state at temperatures below $T_{g}$ and relax over time [47,48]. Therefore, the observed phase-separation kinetics only apply to similarly prepared RIT/PVPVA ASDs. However, it is worth mentioning that water sorption leads to rubbery ASDs where $T_{g}$ is lower than storage temperature [49] (see also Figure 8). We are thus convinced that the subsequently presented kinetics are transferable to other RIT/PVPVA ASDs.

The spatial distributions from the Raman mapping experiments, for example, in Figure 3 and Figure 5, were used to obtain averaged compositions for all phases present. Figure 9a,c shows the time-dependent evolution of these concentrations in the two evolving phases in the 20 wt% DL ASD at 25 °C and 94% RH. As already mentioned, water-sorption kinetics was much faster and was completed before the onset of amorphous phase separation. At 25.2 h (Figure 3a,d,g), all components were homogeneously distributed, and the mass fractions were averaged over the whole map. The following maps (inter alia Figure 3b,e,h or c,f,i) show two-phase systems. Therefore, we averaged the mass fractions of every component at the five RIT-richest points and at the five RIT-poorest points on each map to determine the temporal evolutions of the RIT-rich and RIT-poor phases, respectively. Averaging was based on the observation that the time to generate one map was very short compared to the time for completion of phase separation.

The composition of the emerging RIT-rich phase (Figure 9a) evolves towards the PC-SAFT-predicted equilibrium compositions (earlier work [11] and Figure 8) and the measurement from [11]. However, after 330.2 h, it is still far from the thermodynamic equilibrium composition. In contrast, the composition of the RIT-poor phase (Figure 9c) showed no significant evolution over time. This leads to the conclusion that the kinetics of amorphous phase separation in the RIT/PVPVA ASD with 20 wt% DL at 25 °C and 94% RH is very slow, as no equilibrium was reached within the investigated time frame of 330.2 h.

Similarly, Figure 9b,d illustrate the time-dependent evolution of the compositions in the 25 wt% DL ASD at 25 °C and 94% RH. The first three maps (14.7 h ≤ t ≤ 25.4 h, e.g., see Figure 5a,d,g) showed homogeneously distributed components, and the mass fractions of each component were again averaged by considering all points of the maps. The first three concentrations of each component in Figure 9b,d mainly reflect the kinetics of water sorption prior to phase separation. As weight fractions are reported, an increase in the amount of water leads to decreasing weight fractions of RIT and PVPVA compared to those of the freshly prepared dry ASDs (t = 0 h). Concentration changes after complete water sorption can be attributed to phase separation. For all maps with two distinct phases (t ≥ 30.8 h), the same averaging method of five points similar to the 20 wt% DL ASD was applied to determine the compositions in the RIT-rich phase (Figure 9b) and the RIT-poor phase (Figure 9d) over time.

The compositions in Figure 9b,d reveal a strongly pronounced discontinuity for 25.2 h ≤ t ≤ 30.8 h due to the onset of amorphous phase separation when reaching and exceeding the glass transition according to Figure 8b. While the RIT-poor phase reached its predicted thermodynamic equilibrium composition almost instantaneously, the RIT-rich phase slowly approached the equilibrium predicted and measured in our earlier work [11] and did not converge within 220.1 h. Generally, the compositional changes in the two developing phases have completely different kinetic rates, but both proceed simultaneously until equilibrium is attained. The higher rate of the compositional changes in the emerging RIT-poor phase (Figure 9d) compared to the evolving RIT-rich phase (Figure 9 b) can be explained via the location of the feed point for amorphous phase separation marked in Figure 8. For both ASDs, the feed point is closer to the equilibrium composition of the RIT-poor phase than to the equilibrium composition of the RIT-rich phase (both located at the end points of the tie lines). This means that the emerging RIT-poor phase has to overcome relatively small concentration differences (which is fast) to reach the equilibrium composition, whereas the emerging RIT-rich phase has to overcome huge concentration differences.

Comparing the kinetics of amorphous phase separation in the 20 wt% DL ASD with the one in the 25 wt% DL ASD in Figure 9 shows a vast difference in the corresponding rates. The equilibrium compositions were more closely approached in the 25 wt% DL ASD than in the 20 wt% DL ASD, although the investigated time interval for the 25 wt% DL ASD (220.1 h) was shorter than for the 20 wt% DL ASD (330.2 h). This difference in kinetics can be attributed to the higher overall polymer content in the 20 wt% DL ASD leading to higher viscosities and higher interfacial tensions during phase separation [43]. Thus, both coalescence of droplets and mass transfer across the interface were more hindered in the ASD with lower DL [44]. Thus, the overall polymer content is also responsible for the observation that the compositions are almost equally distributed along the entire tie line (Figure 8) after these long periods of time of 220.1 h or 330.2 h.

However, after an infinite time, only the two equilibrium compositions at the end of the tie lines will remain [11]. The kinetics of amorphous phase separation determined in this work are also in agreement with X-ray fluorescence experiments from Shi et al. [30] at RH → 100% (use of a water drop). Their measurements confirm the completely different kinetics of amorphous phase separations in RIT/PVPVA ASDs with 20 wt% DL and 25 wt% DL. Due to the higher RH, resulting in higher water contents in the ASDs, the onsets of phase separation were earlier, and the kinetics were faster than in this work. The more water is absorbed, the lower is the resulting polymer weight fraction (for quantitative information after storage at 94% RH and 97% RH, see our previous publication [11]). Therefore, the ASDs investigated by Shi et al. [30] had lower viscosities, and the interfacial tensions between the two phases were smaller than for the ASDs considered in this work. Thus, droplets could faster coalesce, and mass could be transferred more quickly across interfaces. The fact that amorphous phase separation is significantly getting faster for DLs higher than 20 wt% was also confirmed by Yang et al. [32], who used confocal fluorescence microscopy to monitor the morphology evolution in RIT/PVPVA ASDs with 20 wt% DL and 30 wt% DL at 37 °C and RH → 100% (use of a water drop).

It has to be mentioned that Shi et al. [30] used approximately 10 min to seal and align the measurement cell. From a thermodynamic point of view, it also requires a particular time after sealing to establish an equilibrium between the liquid phase (water or salt solution), the vapor phase, and the ASD. Thus, the faster the amorphous phase separation, the less accurate the determination of the onset for the corresponding RH. These factors were negligible in this work because the aligned measurement cell was sealed immediately and because the kinetics of water sorption was much faster than the kinetics of amorphous phase separation. Therefore, it was possible to quantify the kinetics of amorphous phase separation separately from the kinetics of water sorption, nearly without superposition.

## 4. Conclusions

In this work, the kinetics of water-induced amorphous phase separation in ASDs and the compositions of the evolving phases were quantified for the first time using Raman spectroscopy. It is safe to say that the results can be transferred from RIT/PVPVA ASDs to other polymer-based ASDs containing APIs with low crystallization tendencies.

Recurrent Raman maps of RIT/PVPVA ASDs with DLs of 20 wt% and 25 wt% at 25 °C and 94% RH enabled identification of the formation of RIT-rich droplets dispersed in a continuous RIT-poor phase. The measured compositions within the phase-separating ASDs were in perfect agreement with the ternary phase diagram of the RIT/PVPVA/water system predicted by PC-SAFT in our previous work [11]. This agreement applied to both the tie-line slope and the tie-line length in the ternary phase diagram. Data from Raman mapping were also consistent with equilibrium compositions previously determined via a newly developed approach using the glass-transition temperatures of phase-separated RIT/PVPVA ASDs [11].

The spatial distributions of RIT, PVPVA, and water in the wet ASD were used to extract the kinetics of amorphous phase separation and the compositions in the two coexisting phases. Generally, the kinetics were very slow for ASDs of both 20 wt% and 25 wt% DL as no thermodynamic equilibrium was attained within 330.2 h and 220.1 h, respectively. The compositional evolution was much slower for the 20 wt% DL ASD than the 25 wt% DL ASD. Even though the investigated time was much longer for the 20 wt% DL ASD, the composition of the RIT-rich phase at the end of the measurement was further away from thermodynamic equilibrium composition compared with the 25 wt% DL ASD. The slower kinetics in the 20 wt% DL ASD was attributed to higher polymer content leading to higher viscosities and higher interfacial tensions during amorphous phase separation. Comparison to data from the literature moreover reveals a significant increase in amorphous phase separation with increasing RH (RH → 100%). This is particularly important when exposing a dry homogeneous RIT/PVPVA ASD to an aqueous dissolution medium. Water from the dissolution medium will be absorbed into the ASD and will induce amorphous phase separation in the ASD, leading to a non-congruent release [11].

## Figures and Tables

**Figure 1 pharmaceutics-15-01395-f001:**
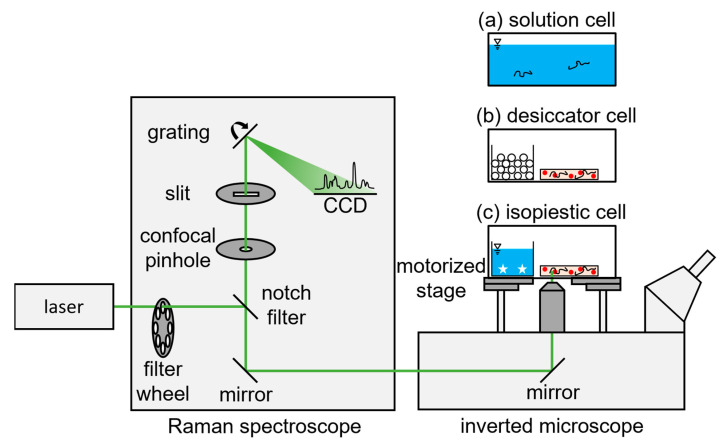
Schematic confocal Raman spectroscope and inverted microscope. The temperature-controlled measurement cell was used as (**a**) solution cell, (**b**) desiccator cell, and (**c**) isopiestic cell.

**Figure 2 pharmaceutics-15-01395-f002:**
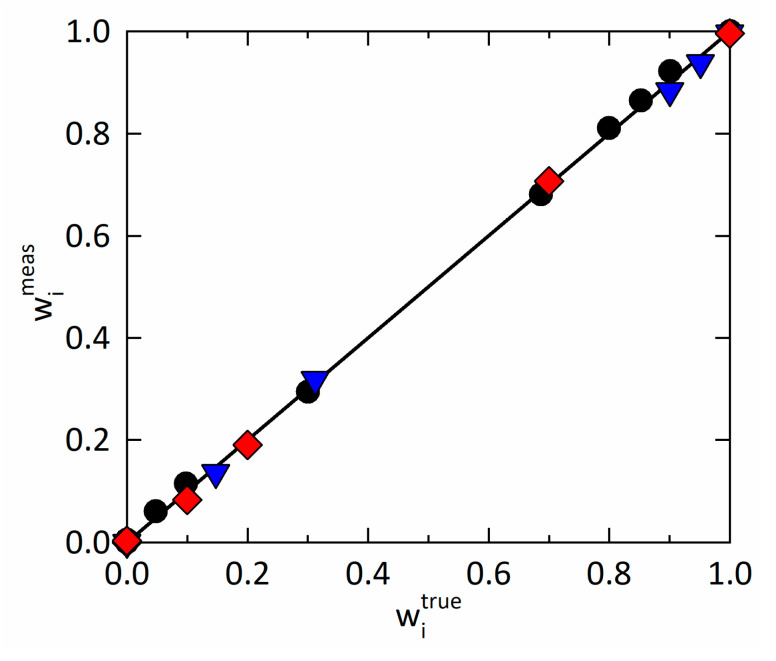
Measured mass fractions $w_{i}^{\mathrm{meas}}$
vs. true mass fractions $w_{i}^{\mathrm{true}}$ for RIT (red diamonds), PVPVA (black circles), and water (blue triangles) at 25 °C. Sample compositions used for calibration are listed in Appendix A.

**Figure 3 pharmaceutics-15-01395-f003:**
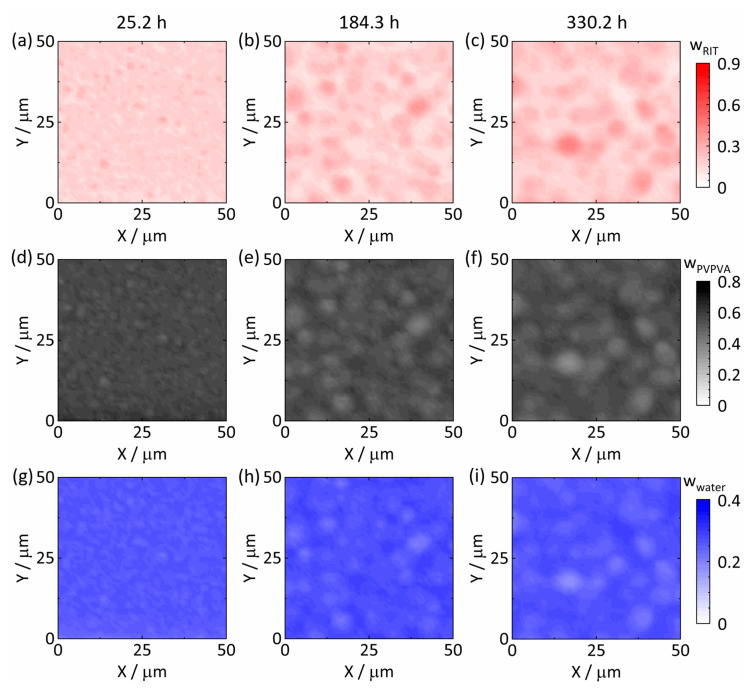
Spatial distributions of RIT, PVPVA, and water for 20 wt% RIT in the dry ASD at 25 °C, 94% RH after 25.2 h ((**a**,**d**,**g**), respectively), 184.3 h ((**b**,**e**,**h**), respectively), and 330.2 h ((**c**,**f**,**i**), respectively). All compositions are given in mass fractions. For a video of the time evolution of the spatial distribution of RIT, the reader is referred to Appendix A.

**Figure 4 pharmaceutics-15-01395-f004:**
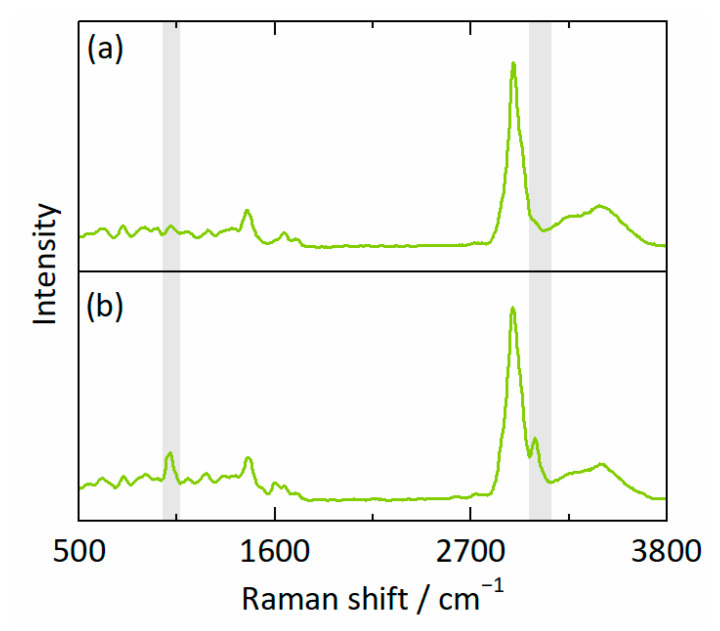
Raman spectra of (**a**) the RIT-poor phase ($w_{\mathrm{RIT}}^{L1}$
= 0.08, $w_{\mathrm{PVPVA}}^{L1}$ = 0.63, and $w_{\mathrm{water}}^{L1}$ = 0.29) and (**b**) the RIT-rich phase ($w_{\mathrm{RIT}}^{L2}$ = 0.46, $w_{\mathrm{PVPVA}}^{L2}$ = 0.36, and $w_{\mathrm{water}}^{L2}$ = 0.18) corresponding to the points X = 33 µm, Y = 34 µm and X = 16 µm, Y = 18 µm for 20 wt% RIT in the dry ASD at 25 °C and 94% RH after 330.2 h (Figure 3c,f,i). Shaded Raman-shift intervals show remarkable contributions of RIT to the mixture spectra.

**Figure 5 pharmaceutics-15-01395-f005:**
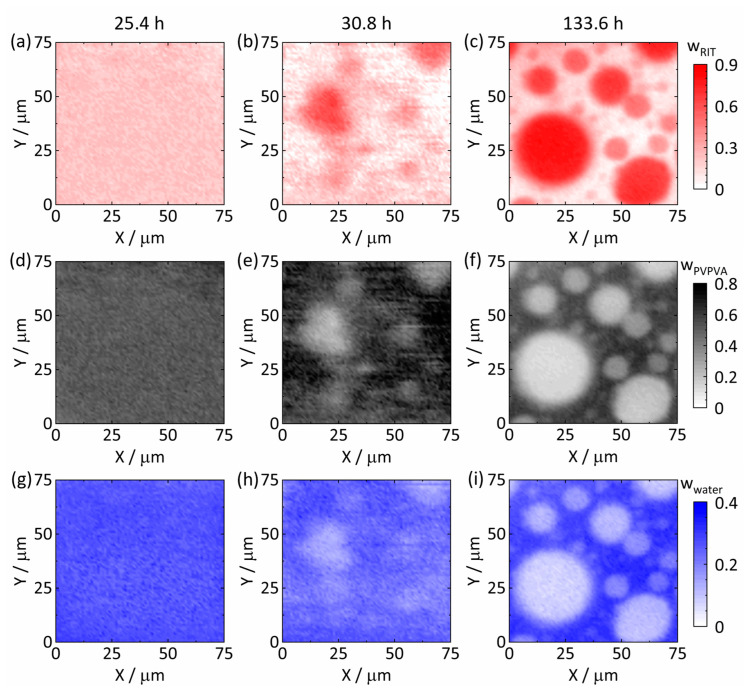
Spatial distributions of RIT, PVPVA, and water for 25 wt% RIT in the dry ASD at 25 °C, 94% RH after 25.4 h ((**a**,**d**,**g**), respectively), 30.8 h ((**b**,**e**,**h**), respectively), and 133.6 h ((**c**,**f**,**i**), respectively). All compositions are given in mass fractions. For a video of the time evolution of the spatial distribution of RIT, the reader is referred to Appendix A.

**Figure 6 pharmaceutics-15-01395-f006:**
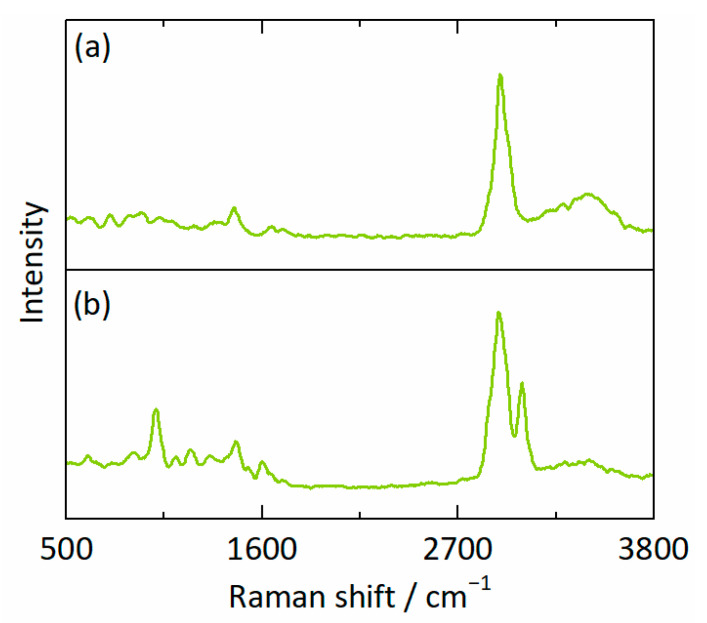
Raman spectra of (**a**) the RIT-poor phase ($w_{\mathrm{RIT}}^{L1}$
= 0.01, $w_{\mathrm{PVPVA}}^{L1}$ = 0.67, and $w_{\mathrm{water}}^{L1}$ = 0.32) and (**b**) the RIT-rich phase ($w_{\mathrm{RIT}}^{L2}$ = 0.83, $w_{\mathrm{PVPVA}}^{L2}$ = 0.11, and $w_{\mathrm{water}}^{L2}$ = 0.06) corresponding to the points X = 51 µm, Y = 34 µm and X = 20 µm, Y = 35 µm for 25 wt% RIT in the dry ASD at 25 °C, 94% RH after 133.6 h (Figure 5c,f,i), respectively.

**Figure 7 pharmaceutics-15-01395-f007:**
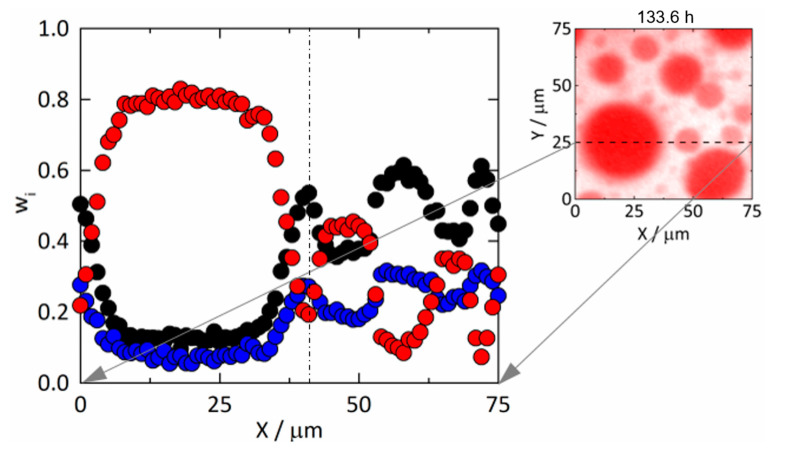
Mass fractions of RIT (red), PVPVA (black), and water (blue) corresponding to the spatial distributions at Y = 25 µm from Figure 5c,f,i in a 25 wt% DL RIT/PVPVA ASD at 25 °C, 94% RH, and 133.6 h. The dash-dotted line indicates X = 42 µm.

**Figure 8 pharmaceutics-15-01395-f008:**
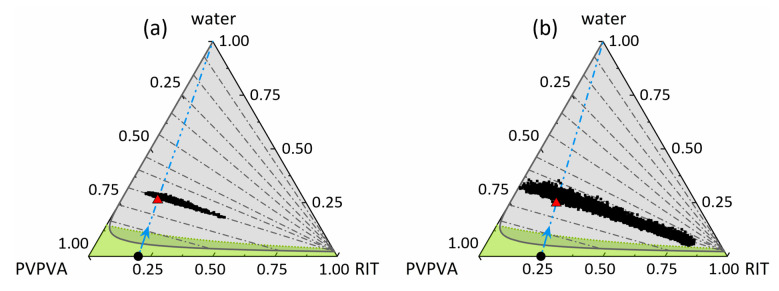
Ternary phase diagram of RIT/PVPVA/water at 25 °C and 0.1 MPa depicting the pathways of water sorption and amorphous phase separation in the ASD with (**a**) 20 wt% DL and (**b**) 25 wt% DL at 94% RH. All compositions are given in mass fractions. The black circles illustrate the dry ASDs, and the blue dash-dotted lines depict the path for water sorption. The gray solid line is the PC-SAFT predicted miscibility gap (gray area), including gray dash-dotted tie lines from an earlier work [11]. The green dotted line is the glass transition calculated using the Kwei equation and enclosing the glassy region (green area) from an earlier study [11]. The red triangle denotes the gravimetrically determined feed point for amorphous phase separation from (**a**) our previous work [11] and (**b**) this work. Black dots are compositions corresponding to the spatial distributions from (**a**) Figure 3c,f,i at 330.2 h and (**b**) Figure 5c,f,i at 133.6 h.

**Figure 9 pharmaceutics-15-01395-f009:**
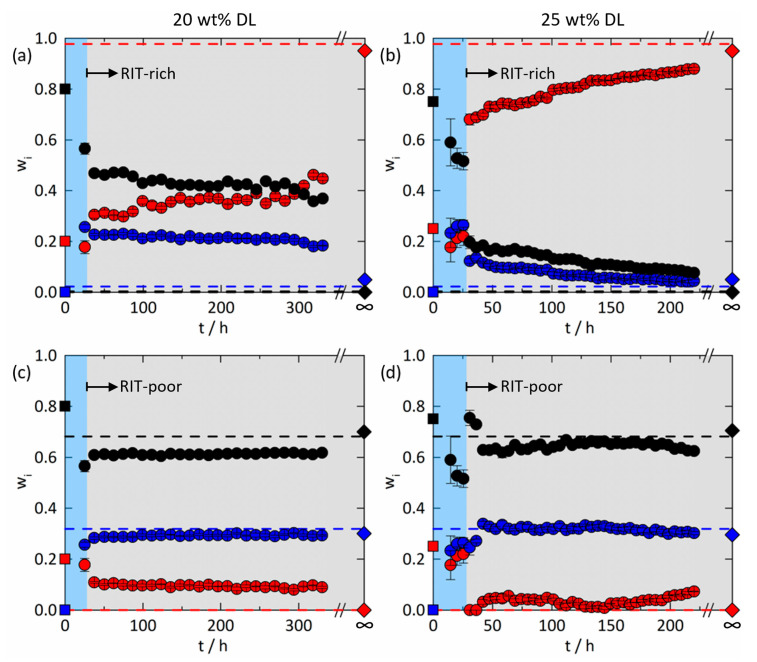
Time-dependent evolution of the compositions in RIT/PVPVA ASDs at 25 °C and 94% RH: The blue region represents the single-phase system during water sorption. The gray region represents the two-phase system after water sorption with the evolution of (**a**) the RIT-rich phase and (**c**) the RIT-poor phase, as well as (**b**) the RIT-rich phase and (**d**) the RIT-poor phase for the 20 wt% and 25 wt% DL ASD, respectively. All compositions are given in mass fractions of RIT (red), PVPVA (black), and water (blue). Squares depict the composition of the freshly prepared dry ASD. Circles are compositions based on Raman mapping, and error bars represent the standard deviations. Diamonds represent equilibrium compositions determined using modulated differential scanning calorimetry in our previous work [11]. Dashed lines are equilibrium compositions predicted with PC-SAFT [11].

## Data Availability

Data is contained within the Article or Appendix A.

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
