# Peer review of "Kinetics of Water-Induced Amorphous Phase Separation in Amorphous Solid Dispersions via Raman Mapping"

_pharmaceutics, 2023, doi:10.3390/pharmaceutics15051395_

Round 1
Reviewer 1 Report
The current research focusing on the kinetics of water-induced phase separation of the amorphous drug is very well designed and executed. The research matches the journal's scope, and the current focus is novel. However, a few of the minor comments are needed to be addressed before the manuscript can be accepted for publication.
1. Please rewrite the sentence in lines 32-34.
2. Since the main focus of the current research is to investigate the kinetics of water-induced phase separation of the amorphous drug, the authors are requested to provide some basic background information about the drug substance, such as melting point, and degradation point.
3. Section 2.2.1. the blend was prepared for a total quantity of 0.7g. What was the drug load? Please specify.
4. Section 2.2.2: Authors mentioned that the samples of VCM were stored at desired conditions. Can the authors please specify the conditions?
5. Section 2.2.4: Authors mentioned that kinetics were measured at fixed RH and temperature. Can authors please specify?
6. Please cross-check the content in line 312.
7. Please condense the content in the conclusion. Please focus on the main observations and outcomes of the current research study.
Reviewer 2 Report
This is an interesting work on using Raman Mapping techniques to determine the phase separation kinetics of amorphous solid dispersions. It is an excellent work in which they use indirect hard modeling to evaluate the changes obtained in the Raman spectra as a function of time. The authors develop a calibration model which is then used to evaluate the prepared dispersions and to monitor the phase separation processes over time.
The models and the treatments are adequate so I consider that it can be accepted for publication in Pharmaceutics.
There are some errors in the references in the figure captions that should be corrected.
Reviewer 3 Report
The authors present a study on quantifying the kinetics of water-induced amorphous phase separation in two amorphous solid dispersions of ritonavir-poly(vinylpyrrolidone-co-vinyl acetate) by Raman spectroscopy, the spectra were evaluated using indirect hard modeling.
The paper presents a good introduction, adequately contextualizes the reader, the knowledge gap is clear and the objectives of the manuscript are concrete.
The materials are well described, and the methodology in particular is very well described.
The experimental quality of the results guarantees the quality of the manuscript, and the analysis of the results allows to understand the results.
Finally, the conclusions correlate with the results.
It is recommended to accept the manuscript after some minor changes.
1. The use of hyperbole is recurrent, "Our previous work provided an in-depth analysis", "first time to investigate" among others. The work is of sufficient quality to be written entirely in the third person without the use of hyperbole.
2. Figure 8 has the potential to be improved, the size of the number labels can be reduced.
3. In figure 9, the time after the bars should be indicated //
4. In the conclusions, it is recommended to remove the phrase "We are convinced", in addition to the fact that the authors are convinced, the important thing is to convince the readers through the quality of the results. The research work must convince by its results.
